# A Diamond/Graphene/Diamond Electrode for Waste Water Treatment

**DOI:** 10.3390/nano13233043

**Published:** 2023-11-29

**Authors:** Yibao Wang, Zhigang Gai, Fengxiang Guo, Mei Zhang, Lili Zhang, Guangsen Xia, Xu Chai, Ying Ren, Xueyu Zhang, Xin Jiang

**Affiliations:** 1Institute of Oceanographic Instrumentation, Qilu University of Technology (Shandong Academy of Sciences), Qingdao 266100, China; yibaowang2021@163.com (Y.W.); guofx1234@163.com (F.G.); zhangm2012@126.com (M.Z.); zllouc@163.com (L.Z.); 13465860124@139.com (G.X.); chaixu211@126.com (X.C.); 2Engineering and Technology Research Center of Diamond Composite Materials of Henan, School of Materials Science and Engineering, Henan University of Technology, Zhengzhou 450001, China; ying_ren@haut.edu.cn; 3Institute of Materials Engineering, University of Siegen, 57076 Siegen, Germany; xin.jiang@uni-siegen.de

**Keywords:** diamond, sandwich structure electrode, water treatment

## Abstract

Boron-doped diamond (BDD) thin film electrodes have great application potential in water treatment. However, the high electrode energy consumption due to high resistance directly limits the application range of existing BDD electrodes. In this paper, the BDD/graphene/BDD (DGD) sandwich structure electrode was prepared, which effectively improved the conductivity of the electrode. Meanwhile, the sandwich electrode can effectively avoid the degradation of electrode performance caused by the large amount of non-diamond carbon introduced by heavy doping, such as the reduction of the electrochemical window and the decrease of physical and chemical stability. The microstructure and composition of the film were characterized by scanning electron microscope (SEM), atomic force microscopy (AFM), Raman spectroscopy, and transmission electron microscopy (TEM). Then, the degradation performance of citric acid (CA), catechol, and tetracycline hydrochloride (TCH) by DGD electrodes was systematically studied by total organic carbon (TOC) and Energy consumption per unit TOC removal (EC_TOC_). Compared with the single BDD electrode, the new DGD electrode improves the mobility of the electrode and reduces the mass transfer resistance by 1/3, showing better water treatment performance. In the process of dealing with Citric acid, the step current of the DGD electrode was 1.35 times that of the BDD electrode, and the energy utilization ratio of the DGD electrode was 2.4 times that of the BDD electrode. The energy consumption per unit TOC removal (EC_TOC_) of the DGD electrode was lower than that of BDD, especially Catechol, which was reduced to 66.9% of BDD. The DGD sandwich electrode, as a new electrode material, has good electrochemical degradation performance and can be used for high-efficiency electrocatalytic degradation of organic pollutants.

## 1. Introduction

As the population grows, the supply of clean water has become a great challenge. The destruction of existing water resources and water recycling systems by human activities has further exacerbated this challenge. Urbanization, industrial expansion, and wars around the world have resulted in the discharge of a considerable amount of wastewater containing high concentrations of trace metals, among other contaminants, into the environment [1,2]. In terms of organic pollutants, in addition to hormones, pesticides, and antibiotics produced by traditional agriculture and medical industries, there are also some emergent pollutants, i.e., pharmaceutical compounds, pharmaceutical and personal care products, endocrine disrupting chemicals, as well as new or opportunistic pathogens, which further increase the pollution of water resources [3,4]. These pollutants not only damage the environment but also pose a serious threat to the health of humans and other biological systems. Take transition elements (such as titanium (Ti), vanadium (V), chromium (Cr), manganese (Mn), iron (Fe), cobalt (Co), nickel (Ni), copper (Cu), and zinc (Zn)), which are the most common in waters contaminated with metals from man-made sources. Research shows that excessive amounts of those metals in water may be precursors to several diseases, such as myocardial infarction, hemochromatosis, and liver cancer caused by excessive and prolonged exposure to Fe, lung cancer caused by hexavalent Cr, and nose cancer caused by Ni. Cd, Cr, Hg, and Pb are highly toxic to biological life and are precursors to several types of cancer [1,5,6,7]. Moreover, organic molecular pollutants, such as catechol, resorcinol, p-nitrophenol, 4-chlorophenol, sulfamethazine, 1H-benzotriazole, carbamazepine, perfluorooctanoate, polyethylene, polypropylene, polystyrene, and nutrients (especially nitrogen and phosphorus), as well as pathogenic microorganisms, have been found to be harmful to human health. It also poses a threat to public health.

The good news is that human beings have also invested a lot of resources in removing pollutants and purifying water and have developed numerous water purification solutions, such as membrane filtration, ion exchange, phytoremediation, physical and chemical adsorption, chemical coagulation, electrocoagulation, advanced oxidation processes (AOPs), i.e., photocatalytic treatments, electro-Fenton oxidation, and hydrodynamic cavitation [8,9,10]. These methods can be used alone or in combination with two or more of them, and they have been widely successful in treating contaminants in water. In recent years, electrochemical treatments have been receiving more interest. The most commonly used technique is electrochemical advanced oxidation processes (EAOPs), which belong to the class of advanced oxidation processes (AOPs), due to their operational simplicity, stability, adaptability, low environmental impact, and high capability for removing organic pollutants from water. When the potential was applied, organic species in water would be oxidized on the anode surface (directly oxidation) or oxidized by in-situ electro-generated species (indirectly oxidation), hydroxyl radicals (·OH), activated chlorine (Cl·), etc. Therefore, the process of making EAOPs depends on the anode material. According to the oxygen evolution potential (OEP), the electrodes can be divided into non-active anodes with a higher OEP and active anodes with a lower OEP. Although the active anodes have strong bonds with hydroxyl radicals and high activity for oxygen evolution reactions (OER), the non-active anodes can reduce undesired OER and support complete oxidation, coupled with their strong stability, and are considered to be more potential electrodes.

The excellent electrochemical performance of boron-doped diamond (BDD) has been studied and applied in many fields, such as heavy metal detection, dopamine detection, electrochemical sensing detection, and so on [11,12,13,14,15]. In terms of environmental governance, BDD electrodes are quite promising anodes for electrochemical treatment of organic wastewater and high-chemical oxygen demand (COD) wastewater because of their extremely wide potential window, remarkable corrosion stability, and low adsorption [16,17,18]. Recently, many papers have reported the application of BDD electrodes in the field of EAOPs and put forward many effective methods to improve the performance of BDD electrodes [19,20]. Most of these works focus on optimizing the surface morphology of BDD electrodes and the three-dimensional structure of metal substrates [titanium, nickel, etc.]. Metal substrates not only improve electrode conductivity but also provide more choices for overall electrode morphology and microstructure. However, at the same time, the composite structure of metal and BDD also brings some problems to the electrode, such as the corrosion of the metal substrate caused by the BDD film falling off or not being completely covered during the deposition process, which decreases the stability of the electrode and could significantly reduce the service life of the electrode [19,20]. Although heavily doped diamond film can directly improve the conductivity of the electrode, it also introduces non-diamond carbon and other problems that greatly reduce the excellent performance of the diamond electrode, such as the reduction of the electrochemical window and physicochemical stability. Therefore, the preparation of BDD electrodes with high stability, high electrochemical oxidation performance, and low energy consumption is the key to expanding their application range.

The combination of diamond and graphene has become a research hotspot [21,22,23,24,25,26,27]. Lan Guojun et al. developed a diamond-graphene material composed of a nanodiamond core and defective graphitic shells that provides high activity in acetylene hydronchlorination [21,22]. In Yuan’s work, few-layer graphene was directly formed on a high-pressure, high-temperature (HPHT) diamond substrate via sp^3^-to-sp^2^ conversion by catalytic thermal treatment and using diamond itself as the carbon source. Because of graphene’s high conductivity, the electrode had a linear electrochemical response to dopamine in a broad range [23]. Li et al. obtained a boron-doped graphene/BDD composite film by chemical vapor deposition. The new electrodes were suitable for the practical detection of markers in human body fluids [24,25]. Bogdannowica R found boron-doped diamond/carbon nanowall films, which provide high electrical conductivity [26]. These works not only demonstrate the excellent electrochemical performance of diamond/graphene composites but also verify the high binding force of the composite electrodes bonded by covalent bonds [27]. However, at present, the microstructure of the BDD/graphene composite electrode is mainly dominated by graphene growing vertically on the diamond surface, which is inevitable to introduce more sp^2^ bonds while improving the conductivity and specific surface area so as to reduce the overall potential window and corrosion resistance of the electrode. In our group’s previous work [28], we proposed an BDD electrode with graphene sandwich structure (named DGD), which can effectively improve electrode conductivity and carrier transfer rate while preserving the excellent intrinsic electrochemical performance of BDD electrodes. That sandwich-structure BDD electrode has potential application in EAOPs.

In this paper, BDD self-supporting electrodes with different sandwich structures were prepared. The prepared electrodes were then employed to degrade Citric acid (CA), Catechol and Tetracycline hydrochloride (TCH). As a common additive, CA produces a large amount of organic waste liquid during food processing. Catechol is an important chemical intermediate that was identified as a Group 2B carcinogen in 2017. And TCH is a widely used antibiotic, but its water-soluble properties also cause severe pollution risks for the environment. We investigated the total organic carbon (TOC) and Energy consumption per unit TOC removal (EC_TOC_) for different organic solutions by DGD electrodes with different graphene layer thicknesses. It was found that the appropriate thickness of the graphene layer can not only improve the degradation rate of organic pollutants but also reduce the energy consumption of degradation. Combined with the microstructure analysis of the films, the effect of the interlayer structure and graphene interlayer on the degradation performance of the electrodes was discussed. This work combines the properties of BDD with graphene, making a novel and commercially viable DGD electrode for the electrocatalytic degradation of organic pollutants. Figure 1 illustrates the schematic of the preparation process for the DGD sample.

## 2. Experimental Details

### 2.1. Preparation of the BDD and DGD Electrodes

The BDD films (500 µm) were prepared using a hot filament chemical vapor deposition system (HFCVD).The deposition conditions were as follows: chamber pressure was 3–5 kPa, hot wire temperature was 2400–2500 °C, base temperature was 850 °C, doping concentration was 4000 ppm, and growth rate was 2.5 μm/h. The graphene layer was prepared by copper-metal-assisted vacuum annealing. The preparation process of the upper surface BDD was consistent with that of the bottom layer BDD, and the thickness was 800 nm. The detailed preparation parameters for Cu deposition and vacuum annealing are listed in Appendix A. The DGD electrode is named DG_x_D (x is the time of vacuum annealing).

### 2.2. Characterization

A Raman spectrometer with a laser excitation wavelength of 532 nm was used to measure the characteristics of the as-prepared electrodes. Field-emission scanning electron microscopes were used to assess the morphology of the sample surface, and energy-dispersive X-ray spectrometry (EDS SU3800, Tokyo, Japan) was used to reveal the chemical composition. Transmission electron microscopy (TEM) (talos f200s, 200Kev, Waltham, MA, USA) was used to analyze the microstructure of samples. The TEM samples were prepared by FIB (FEI Nova 450, Portland, OR, USA). Surface chemical bonds were recorded by XPS using Al Kα as the X-ray source. The electrochemical performance of the developed electrodes at room temperature was tested using the CHI600E electrochemical workstation (Shanghai Chen-hua Inc., Shanghai, China). BDD electrodes (exposed geometric area of 0.28 cm^2^), platinum sheet, and Ag/AgCl (sat. KCl) electrodes were used as the working electrode, counter electrode, and reference electrode, respectively. The electrochemical windows were tested by cyclic voltammetry tests in 0.1 M H_2_SO_4_ with a scan rate of 100 mV/s. Nyquist plots were recorded by electrochemical impedance spectroscopy (EIS) tests in 0.1 M NaSO_4_ containing 1 mM [Fe(CN)_6_]^3−/4−^ in the frequency range 0.01–100 kHz. The electrocatalytic oxidation potentials for CA and alcohol were determined by current density-time curves at 3 V in 0.1 M Na_2_SO_4_. Water treatment experiment.

Three organic compounds (CA, Catechol, and TCH) were used for degradation at concentrations of 100 mg/L. The supporting electrolyte was 0.1 M Na_2_SO_4_. The PH in the process of electrolysis is not a determined value, and its range is between 7 and 8.

The developed BDD/DGD electrodes and Ti were used as the anode and cathode in the degradation process, respectively, to degrade 25 mL of organic wastewater at room temperature. The electrode size was 15 mm × 10 mm × 1 mm, and the effective area when degraded was 10 mm × 10 mm. The distance between the electrodes (cathode and anode) in the degradation system was 3 mm. Electrolysis liquid was stirred by magnetic agitation. The current density during electrolysis is set to 200 mA/cm^2^. Because of the different structures of electrode materials, the electrolytic voltage range is 6.5 v to 7.5 v. Different pollutants performed ten repeated degradation tests, and the test results were the mean, which served as the basis for discussion.

The TOC was analyzed by the TOC analyzer. The pollutant removal was calculated as follows:TOC Removal (%) = [(C_0_ − C_t_)/C_0_] × 100(1)

Energy consumption per unit of TOC removal (EC_TOC_) was calculated as
(2)$$EC_{TOC}(kWh\cdot g^{-1})=\frac{EIt}{V_{S}△{\left|TOC\right|}_{exp}}$$
where *E* is the voltage (V) in the degradation process, *I* is the current intensity (A), *t* is the degradation time (h), and *Vs* is the volume (m^3^) of solution. $△{\left|TOC\right|}_{exp}$ is the value of the initial TOC minus the TOC at time t during the degradation experiment [19].

## 3. Result and Discussion

### 3.1. Material Characterization

Figure 2 shows the morphologies of the as-prepared BDD and samples treated by metal-assisted vacuum annealing at different times. The single BDD film contains a lot of diamond crystal, which is packed tightly together with a grain size of about 500 nm (Figure 2a). Figure 2b shows the surface of BDD covered with a copper layer of about 1 μm where the clear diamond crystal morphology has disappeared and is replaced by a uniformly covered metal layer. Figure 2c–f exhibits the SEM images of the samples after vacuum annealing at 1000 °C for 10, 20, 30, and 40 min, respectively. It is difficult to distinguish the graphene phase on the surface of the sample with an annealing time of 10 min as shown in Figure 2c. When the annealing time is increased to 20 min (Figure 2d), the surface of the film changes obviously with the appearance of the graphene layer, and the grains are uniformly covered by graphene. As the holding time exceeded 30 min, the thickness and density of some graphene layers continued to increase, as shown in Figure 2e. However, at this time, the graphene distribution is not uniform, and it accumulates in the low area of the diamond surface grain, showing a trend of vertically dense growth. When the annealing time is 40 min, with the increase of graphene layers, the diamond film has been completely covered, and the morphologic characteristics of polycrystalline diamond disappear completely (Figure 2f). Appendix A shows the SEM images of DGD films after the growth of the upper BDD film. The growth processes of the upper and lower BDD films were kept identical, and the surface morphology of Appendix A is almost the same as Figure 2a.

### 3.2. Raman Spectroscopy

Raman is an effective and convenient detection method for characterizing diamonds and graphene. Figure 3a shows the Raman spectra of the as-prepared BDD samples treated by metal-assisted vacuum annealing at different times. For the BDD curve in Figure 3a, 1200 cm^−1^ and 1500 cm^−1^ peaks can be seen, and the 1316 cm^−1^ peak is intense; this is a typical Raman spectrum of heavily doped diamond [19,20,29,30]. When the annealing time comes to 10 min, the curve of BDD-10 min has no obvious change, which indicates that the carbon atoms still remain mainly sp^3^ hybridized, and graphene is not formed after such a short annealing time. As the annealing time reached 20 min, the Raman spectra showed a G peak (1580 cm^−1^) and a 2D peak (2700 cm^−1^), indicating the generation of graphene on the surface of diamond. The ratio of the peak intensities offers another piece of evidence for the structural change as a function of annealing time. In this work, the G peak and 2D peak of the samples are simulated by Lorentz fitting. As shown in Figure 3b, the I_2D_/I_G_ value of BDD-20 min is about 0.42, and the value of full width at half (FWHM) of the 2D peak is 45 cm^−1^, which are typical Raman characteristics of multilayer graphene [31,32,33,34,35]. When the annealing time was extended to 30 min, the 2D peak decreased obviously, indicating that the number of graphite layers increased. However, when the annealing time reached 40 min, amorphous phase transformation occurred on the surface of the BDD electrode, and the 2D peak formed a peak pack and almost disappeared completely. Raman test results show that multiple layers of graphene can be formed on the surface of BDD film with an appropriate annealing time (20 min), which is consistent with the scanning results in Figure 2.

Figure 4a–e show the C 1s XPS map of the BDD film surface at different annealing times. After peak fitting, characteristic peaks located at 285.5 eV and 284.5 eV appear in the XPS spectrum, which correspond to sp^3^-bond carbon in tetrahedral configuration and sp^2^-bond carbon in planar triangle configuration, respectively [36,37,38]. Figure 4f summarizes the ratio of sp^3^ bond carbon to sp^2^ bond carbon. BDD~BDD-30 min film is mainly composed of sp^3^-bond carbon. The ratio of sp^3^/sp^2^ decreases with an increase in annealing time. When the annealing time increases to 40 min, the proportion of sp^3^-bond carbon is only 49%, indicating that there are more graphite phases on the BDD-40 min surface. XPS results show that the carbon content of the sp^2^ bond in the films increases with the increase in annealing time, which is consistent with the Raman spectra results.

Based on the above test results, it is considered that the annealing time of 20 min is the best experimental condition for producing graphene layers. Therefore, the top BDD layers were grown on the surface of the films after vacuum annealing for 10 min, 20 min, and 30 min (DG_10_D, DG_20_D, and DG_30_D) for subsequent study. TEM was performed on the DGD films to unravel the unique microstructure (Figure 5). As shown in the transmission bright field phase, the thickness of the intermediate transition layer increases with the increase in annealing time. The interlayer thickness of DG_10_D, DG_20_D, and DG_30_D samples is 7.3, 21.2, and 27.5 nm, respectively. When the annealing time was 10 min, the thickness of the transition layer was only about 7.3 nm, and the transition layer showed an amorphous structure without the lamellar structure of graphene. Compared with the transformation of diamond during the annealing process, it is considered that this part of amorphous carbon originates from the initial growth stage of the upper BDD layer. As shown in Figure 5b, the cross-section of sample DG_20_D shows obvious characteristics of lamellar graphite, which could be identified by the interplanar distance of 0.38 nm, in agreement with the theoretical values of graphite (002) plane spacing. In addition, the layered graphite is distributed parallel to the BDD layer at the contact with the diamond layer. When the graphene layer extends and finally covers the entire diamond surface during growth, the catalytic reaction terminates because the diamond surface is isolated from the Cu catalyst by the graphene itself [23]. The TEM of DG_30_D shows randomly oriented graphite without a preferred orientation. This result is consistent with our previous published paper.

To examine the electrocatalytic degradation capacity of these BDD electrodes, current density-time curves are recorded in 0.1 M Na_2_SO_4_ with and without 50 mgL^−1^ CA, as shown in Figure 6 The test applied potential is 3 V. Without adding CA, the response current densities of BDD, DG_10_D, DG_20_D, and DG_30_D electrodes are 18.2, 11.2, 7.2, and 7.3 mA cm^−2^, respectively. The current density of BDD electrodes is obviously higher than that of DGD electrodes, which is determined by the oxygen evolution capacity of different electrodes. The oxygen evolution potential (OEP) of the as-prepared electrodes was compared by CV testing (Appendix A). The OEP of DG_20_D reached 2.37 v, higher than that of BDD (2.1 v). Moreover, the window potential of DGD electrodes was similar to that of BDD electrodes, indicating that the DGD structure could improve the OEP of the electrode while retaining the performance of BDD. After adding CA, the response current densities of BDD, DG_10_D, DG_20_D, and DG_30_D electrodes increased to 23.9, 15.3, 16.9, and 13.8 mA cm^−2^, respectively. This leads to step current densities of 5.7, 4.1, 7.7, and 6.5 mA cm^−2^ for BDD, DG_10_D, DG_20_D, and DG_30_D electrodes, respectively. The step current density of the DG_20_D electrode is significantly larger than that of the BDD. The addition of CA increases the response current density of the electrode, which can be attributed to the oxidation of CA. This suggests a greater potential for sandwich-structured electrodes for electrocatalytic degradation of CA. Table 1 lists the energy consumption comparisons of different electrodes during CA degradation in detail. Before CA was added (0–300 s), the hydrolysis energy consumption (E_1_) of the BDD electrode was the highest, while that of the DG_20_D electrode was the lowest. After adding CA, the total energy consumption (E_2_) includes two parts: energy consumed by hydrolysis (E_1_) and energy consumed by electrochemical decomposition CA (E_3_). Therefore, q = (E_3_/E_2_) × 100% is defined as the energy efficiency of the electrode for CA degradation. As can be seen, the q of the DG_20_D electrode is 57.34%, 1.4 times that of the BDD electrode (23.72%).

Three representative organic compounds, carbon chain compounds (CA), single cyclic aromatic hydrocarbons (Catechol), and polycyclic aromatic hydrocarbons (TCH), were selected to comprehensively evaluate the degradation ability of the four electrode materials (BDD, DG_10_D, DG_20_D, and DG_30_D). Figure 7a shows the CA removal of different electrodes. The treatment efficiency of DGD electrodes is superior to that of single-layer BDD electrodes, and the improvement of the DG_20_D electrode is the most obvious. After 20 min, the TOC removal rate of the DG_20_D was close to 80%, while that of the BDD was close to 50%. At other measured time nodes, the TOC removal rate of the DG_20_D electrode was about 20% higher than that of the BDD electrode. In the process of Catechol degradation (Figure 7b), the TOC of the DG_20_D removal rate reached 60% after 20 min, while that of BDD was only 20%. In the degradation process of TCH (Figure 7c), the TOC curves of the four electrodes were similar. Although the degradation rate of the DG_20_D electrode was slightly (about 10%) higher than that of the BDD electrode in the first 40 min, the TOC values of the four electrodes were almost the same after 60 min. Compared with the other two organic compounds, tetracycline hydrochloride is composed of many carbon rings, and its degradation process is very complex, which can produce 10 intermediate products, which may be present in the liquid as by-products if the degradation is not complete [39], as shown in Appendix A. Figure 7d shows the EC_TOC_ of different electrodes for three compounds calculated according to Equation (2). The energy consumption of the electrodes varied with the organic compounds. The average EC_TOC_ of the four electrodes is 1.57 kWh/g, 0.95 kWh/g, and 1.04 kWh/g for the degradation of CA, Catechol, and TCH, respectively. Overall, for these three organic compounds, the EC_TOC_ of the DG_20_D electrode is lower than that of the other electrodes, while the EC_TOC_ of the BDD electrode is the highest. After 60 min of degradation for CA, Catechol, and TCH, the EC_TOC_ of the DG_20_D electrode was 78%, 66.9%, and 81.9% of that of the BDD electrode, respectively. Compared with the BDD electrode, the EC_TOC_ of the DG_10_D and DG_30_D electrodes decreased slightly, especially the EC_TOC_ of the DG_10_D electrode, which was almost the same as that of the BDD electrode.

Therefore, although the addition of a graphene layer does help to improve the degradation rate of the electrode, the thickness of the graphene layer is very important. When the annealing time is too short (less than 10 min in this paper), although the BDD surface can form graphene, it exists in the form of islands and cannot form a continuous graphene phase structure. On the other hand, when the annealing time is too long, the graphene layer turns into graphite. In both cases, it is not conducive to improving the performance of the electrode. Herein, the DG_20_D sandwich electrode prepared with 20 min vacuum annealing had the best structure and significantly improved the degradation rate of organic matter. It should be especially pointed out that the sandwich structure electrode proposed in this paper is still a two-dimensional electrode; this structure is to improve the performance of the BDD electrode itself, and it has the potential to further significantly improve the electrode performance through three-dimensional structure modification, as mentioned in the literature [40].

## 4. Conclusions

In conclusion, copper films were used as catalysts on boron-doped diamonds to successfully induce multilayer graphene, and the formation process of graphene was discussed. Compared to the conventional BDD electrode, the developed DGD sandwich-structure electrode showed better electrochemical degradation performance. In the process of dealing with CA, the step current of the DGD electrode was 1.35 times that of the BDD electrode, and the energy utilization ratio of the DGD electrode was 2.4 times that of the BDD electrode. The energy consumption per unit TOC removal (EC_TOC_) of the DGD electrode was lower than that of BDD, especially Catechol, which was reduced to 66.9% of BDD. So diamond and graphene sandwich electrodes will be a novel and commercially applicable electrode material for efficient electrocatalytic degradation of organic pollutants. This study provides new inspiration for the design of BDD anodes for high-performance EAOPs.

## Figures and Tables

**Figure 1 nanomaterials-13-03043-f001:**
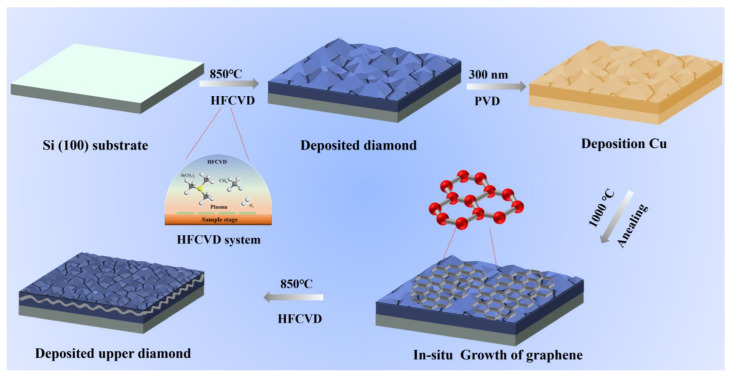
Schematic illustration of the DGD sample preparation process.

**Figure 2 nanomaterials-13-03043-f002:**
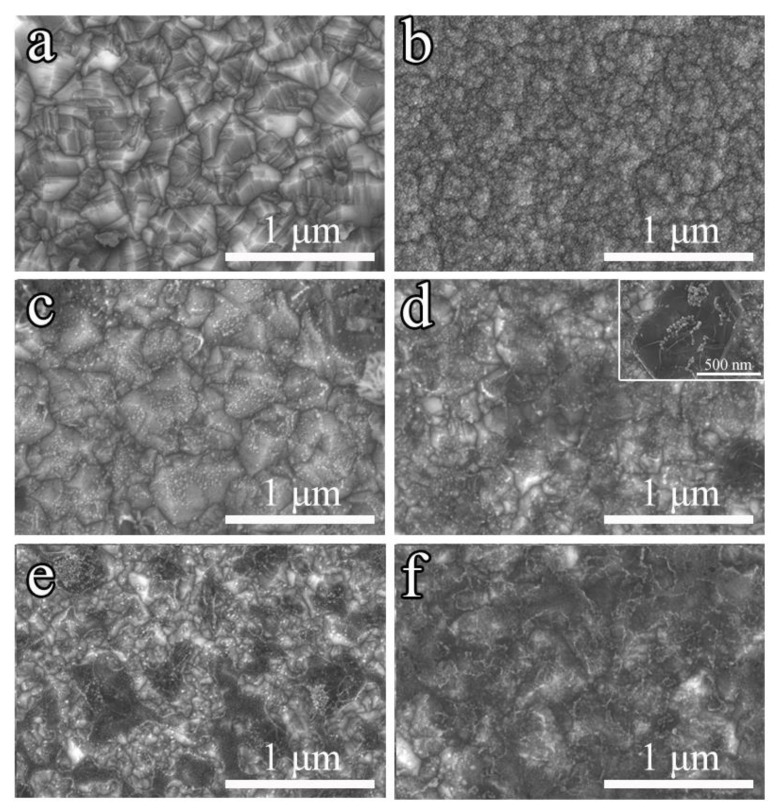
SEM micrograph of diamond; (**a**) image of pure diamond; (**b**) image of films after deposited Cu layers; (**c**) holding 10 min; (**d**) holding 20 min; (**e**) holding 30 min; and (**f**) holding 40 min.

**Figure 3 nanomaterials-13-03043-f003:**
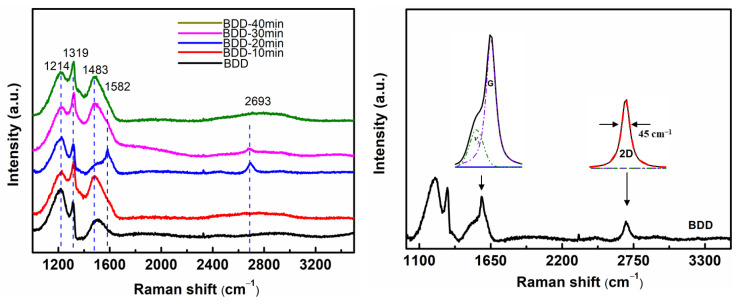
(**a**) Raman spectra of BDD annealed at 1000 °C for different times; (**b**) Image of the Lorentzian profile fitted to the G peak and the 2D peak after annealing for 20 min.

**Figure 4 nanomaterials-13-03043-f004:**
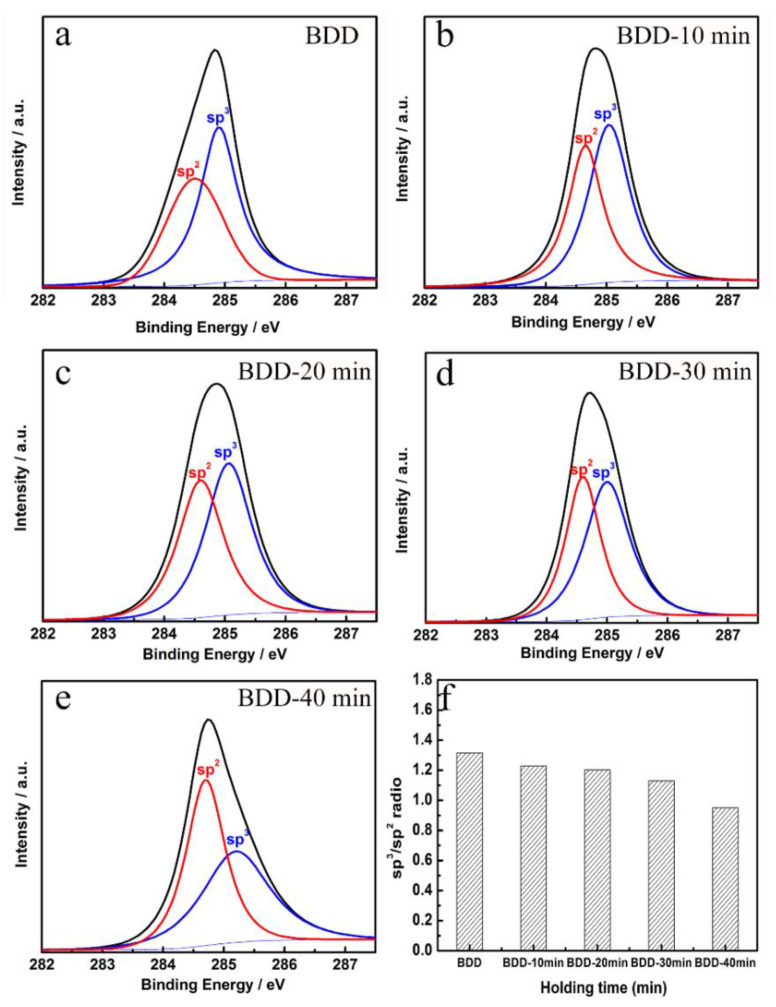
High-resolution C 1s XPS spectra of BDD annealed at 1000 °C for different times: (**a**) BDD, (**b**) BDD-10 min, (**c**) BDD-30 min, (**d**) BDD-50 min, (**e**) BDD-70 min; (**f**) The ratios of chemical species estimated from the deconvoluted C 1s peak of films.

**Figure 5 nanomaterials-13-03043-f005:**
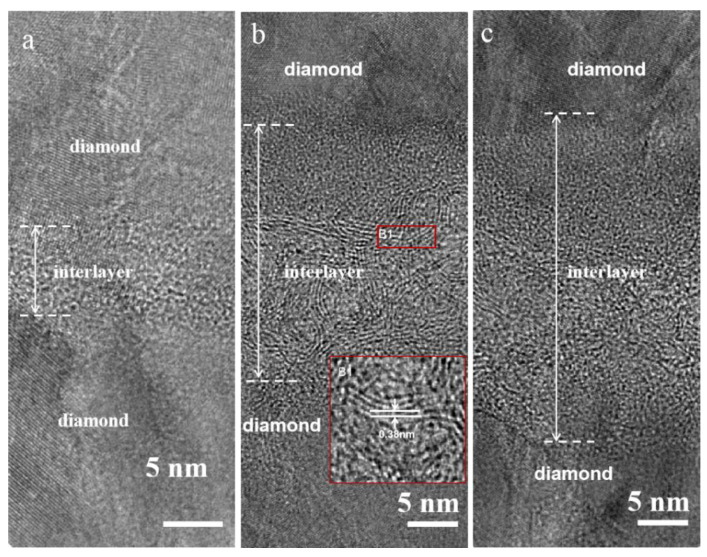
High-resolution TEM image of DGD: (**a**) DG_10_D, (**b**) DG_20_D, and (**c**) DG_30_D.

**Figure 6 nanomaterials-13-03043-f006:**
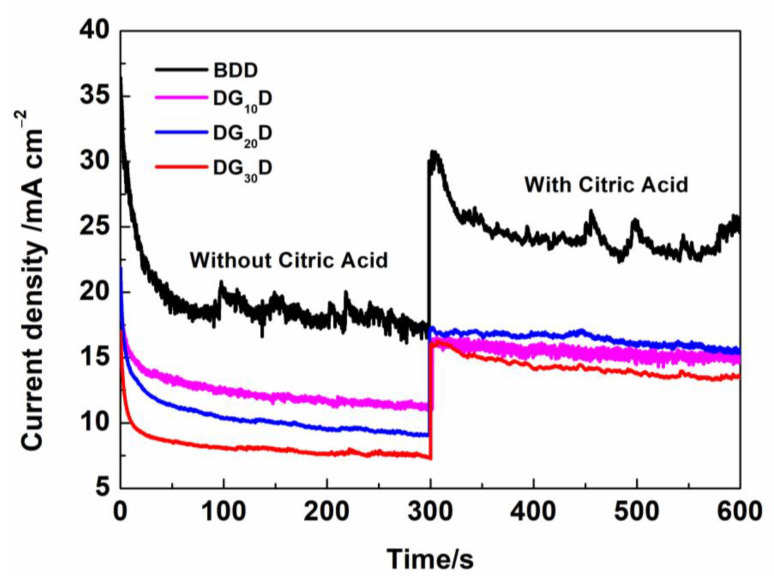
Current density-time curves in 0.1 M Na_2_SO_4_ with and without 50 mg L^−1^ Critric acid.

**Figure 7 nanomaterials-13-03043-f007:**
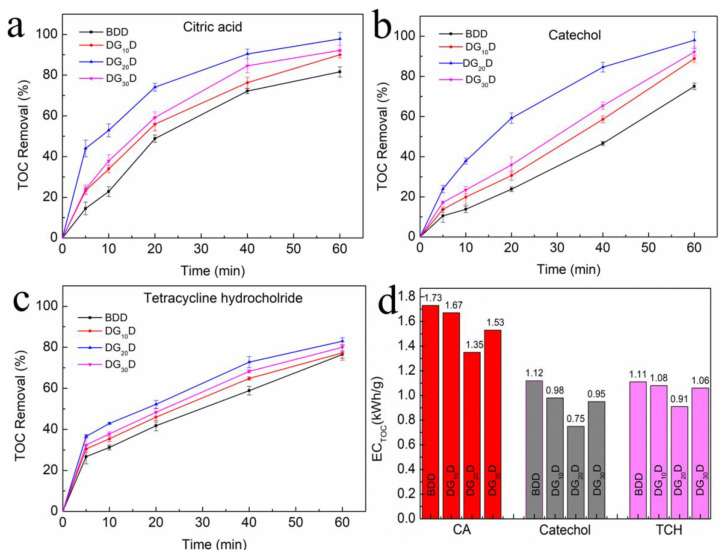
The TOC removal of organics (100 mg/L) as a function of degradation time with the four electrodes. (**a**) Citric acid; (**b**) Catechol; (**c**) TCH; (**d**) Energy consumption per unit TOC removal of electrodes at 60 min.

**Table 1 nanomaterials-13-03043-t001:** Comparison of energy consumption of different electrodes *.

Film Type	E_1_ (w)	E_2_ (w)	E_3_ (w)	Q
BDD	4.62	6.07	1.44	23.72%
DG_10_D	2.84	3.89	1.04	26.73%
DG_20_D	1.83	4.29	2.46	57.34%
DG_30_D	1.85	3.51	1.65	47.00%

* E = iSvt, i, average current density; S, effective area of the electrode; v, voltage between the electrodes; E_1_, 0–300 s energy consumption; E_2_, 300–600 s energy consumption; E_3_ = E_2_ − E_1_; q = (E_3_/E_2_) × 100% efficiency of electric energy.

## Data Availability

Data are contained within the article and Appendix A.

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
