# Peer review of "A Diamond/Graphene/Diamond Electrode for Waste Water Treatment"

_nanomaterials, 2023, doi:10.3390/nano13233043_

Round 1

Reviewer 1 Report

Comments and Suggestions for Authors

The paper submitted by Prof. Xueyu Zhang and co-authors titled “A diamond/graphene/diamond electrode for waste water treatment”.

 The authors suggested a new electrode system for better wastewater treatment. The suggested electrode system adopts the BDD/graphene/BDD (DGD) sandwich structure based on the HFCVD and PVD process controlling the annealing time. When the annealing time is too short (less than 10 minutes), the BDD surface can form graphene in the form of islands. However, when the annealing time is too long, the graphene layer can turn into graphite.

Therefore, the authors are insisting that 20 minutes of annealing time was the best recipe for the DGD electrode system.

 However, I wonder author may misled the result by not interpreting the capacitance increase. As shown in Fig 6, in the absence of reactants (citric acid or alcohol), the BD20D electrode system showed a high background current which implies that the electrode system has high capacitance.

 Electrolysis of the reactants (citric acid or alcohol) can be shown by the current change upon the injection of the reactant. The current change seems similar in Figure 6a and Figure 6b.

 I think the authors can show the effect of reactant electrolysis using the consecutive cyclic voltammetry technique where the authors can neglect the non-Faradaic current.

 Please, add error bars in Figure 7 a,b,c.

In Figure 7b, the current changing trend is different from others. I’m curious if the trend is reproducible.

Comments on the Quality of English Language

Reviewer 2 Report

Comments and Suggestions for Authors

General comment:

The manuscript deals with water treatment by using a boron-doped diamond/graphene/boron-doped diamond sandwich structure electrode. Different sandwich structures were prepared and tested. The characterization of the material was carried out.

The manuscript is suitable to be published in this journal, however, some points should be addressed before publication.

Some minor language mistakes are present that should anyway be corrected.

1. Introduction

Please, improve the literature overview on water treatment by using advanced oxidation processes. Please consider the following manuscripts:

o   An innovative approach for atrazine electrochemical oxidation modelling: Process parameter effect, intermediate formation and kinetic constant assessment (2023) Chemical Engineering Journal, 474, art. no. 146022.

o   Combining ozone with UV and H2O2 for the degradation of micropollutants from different origins: lab-scale analysis and optimization (2019) Environmental Technology (United Kingdom), 40 (28), pp. 3773-3782.

o   Improving degradation of real wastewaters with self-heating magnetic nanocatalysts (2021) Journal of Cleaner Production, 308, art. no. 127385.

2. Experimental details

Please, specify the parameters investigated and their variation range.

Please, specify if by-product formation was investigated.

Please, clarify if investigations were carried out in duplicate/triplicate etc.

Please, clarify if the pH was monitored and/or regulated.

3. Results and discussion

Please, include statistical analysis.

By-products formation should be investigated, and a reaction mechanism proposed.

Please, improve the comparison between your findings and literature data.

Reviewer 3 Report

Comments and Suggestions for Authors

The manuscript is aims to present a novel BDD/graphene/BDD (DGD) sandwich structure electrode, which improves the conductivity.

 i) Introduction, first paragraph, beginning

Where is written "The excellent electrochemical performance of BDD has been studied and applied in many fields [1-5]" it should be "The excellent electrochemical performance of Boron-doped diamond (BDD) has been studied and applied in many fields [1-5]". The same for other acronyms and some variables. They must be described when they first appear.

“…applied in many fields [1-5]". “Many fields” should be detailed. What fields?

 ii) Introduction, between first and second paragraph

BDD electrodes are not only useful in electrocoagulation for COD removal (i.e., for organics removal). They are efficient for removing nutrients (N and P), heavy metals and even pathogens. The introduction practically only talks about the production of the BDD electrode, which I understand taking in account the nature of the work but lacks information about its usefulness in terms of removing pollutants. This information would attract more researchers in reading your work (i.e., it needs some information on the practical application of your electrodes).

Here are some bibliographic suggestions published in MDPI journals, which include applications in water treatment:

1) https://doi.org/10.3390/su15021492

2) https://doi.org/10.3390/su15021708

iii) Section “2.3. Water treatment experiment”

This section presents the performance of an experimental test to evaluate the removal of citric acid, catechol and TCH, the results of which are analyzed in section 3.2 (Figure 7).

The abstract says nothing about this experiment and its conclusions. In the introduction nothing is mentioned about the practical application of the electrode (hence my previous bibliographical suggestions).

In the conclusions they talk about the results of this experiment, but not in the abstract, introduction and objectives.

The text needs to be improved.

Round 2

Reviewer 1 Report

Comments and Suggestions for Authors

Dear authors,

I carefully read the author's response and I come up with more questions.

About the electrochemical data in Figure 6, the data obtained from 'without citric acid' and from 'without alcohol' should be identical because the experimental condition is exactly the same.

However, there are differences in the two data. 

In Figure 6a, DG30D showed the lowest E1 value among the three. 

In Figure 6b, DG20D showed the lowest E1 value among the three. 

The authors should explain this.

One more thing,

In the aspect of alcohol decomposition, it is hard to tell that DG20D showed the best performance.

Comments on the Quality of English Language

okay

Reviewer 2 Report

Comments and Suggestions for Authors

The authors revised the manuscript according to the comments/changes suggested. The paper is suitable to be published in this journal in the current form.
